# Impact of Ovarian Suspensory Ligament Rupture on Surgical Stress in Elective Ovariohysterectomy in Bitches

**DOI:** 10.3390/vetsci11120658

**Published:** 2024-12-16

**Authors:** Pauline Silva dos Santos, Victor Mendes de Oliveira, Keli Cristina Corbellini Oltramari, Vitória Santos Guimarães, Sarah Fernandes, Carla Eduarda dos Santos Ferreira, Agatha Costa Malinski, Vinícius Cardoso de Oliveira, Jéssica Corrêa, Izabelle Moutinho, Dalila Moter Benvegnú, Camila Dalmolin, Gabrielle Coelho Freitas, Fabíola Dalmolin

**Affiliations:** 1Programa de Pós-Graduação em Saúde, Bem-Estar e Produção Animal Sustentável na Fronteira Sul (PPG-SBPAS), Universidade Federal da Fronteira Sul (UFFS), Realeza 85770-000, Brazil; paulinesilvadossantos@gmail.com (P.S.d.S.); kellicristyny@gmail.com (K.C.C.O.); dalila.benvegnu@uffs.edu.br (D.M.B.); gabrielle.freitas@ufsm.br (G.C.F.); 2Programa de Residência em Área Profissional da Saúde em Medicina Veterinária, Universidade Federal do Paraná (UFPR), Palotina 85950-000, Brazil; mendesvictormo@gmail.com (V.M.d.O.); izaah.moutinho@hotmail.com (I.M.); 3Curso de Medicina Veterinária, Universidade Federal da Fronteira Sul (UFFS), Realeza 85770-000, Brazil; visguimaraes07@gmail.com (V.S.G.); sarahf_gh@hotmail.com (S.F.); agathamalinski48@gmail.com (A.C.M.); 4Programa de Residência em Área Profissional da Saúde em Medicina Veterinária, Universidade Federal de Santa Maria (UFSM), Santa Maria 97105-900, Brazil; carlaeduarda.ferreira@outlook.com; 5Residência Multiprofissional em Clínica Cirúrgica de Animais de Companhia, Universidade Federal do Mato Grosso (UFMT), Cuiabá 78060-900, Brazil; mvviniciuscdso@gmail.com; 6Programa de Residência em Clínica Médica de Pequenos Animais, Universidade Estadual de Santa Catarina (UDESC), Lages 88520-000, Brazil; jessica98_correa@outlook.com; 7Departamento de Medicina Veterinária, Centro Universitário Mater Dei (UNIMATER), Pato Branco 85501-200, Brazil; camidal@gmail.com

**Keywords:** pain management, hemostasis, leukocyte response, oxidative stress markers

## Abstract

There is evidence indicating that rupturing the ovarian suspensory ligament (OSL) in bitches—a common step during open ovariohysterectomy (OVH)—may provoke adverse effects, even though it facilitates hemostasis of the ovarian arteriovenous complex. The response to surgical stress may compromise patient recovery and should ideally be minimized. This study evaluated the effects of OSL rupture on the surgical stress response in healthy bitches, aiming to promote faster recovery. Clinical and laboratory assessments revealed that OSL rupture provokes greater hemostatic changes compared to techniques that preserve the ligament.

## 1. Introduction

Ovariohysterectomy (OVH) is one of the most commonly performed surgeries in dogs, with its benefits and potential complications being well documented [1,2]. Several techniques have been described, differing only slightly in their approaches [1,3,4,5].

In young, healthy patients, OVH can be performed through a small ventral midline incision. This limited approach makes exposing the ovarian vessels for ligation one of the most challenging aspects of the surgery. Proper exposure is crucial to ensure secure ligation and prevent any ovarian remnant. In dogs, rupturing the ovarian suspensory ligament (OSL) is commonly recommended to facilitate vessel exposure [3]. However, some evidence suggests that rupturing the ligament may increase the postoperative pain intensity and extend the rupture to the peritoneal attachment, thereby exposing the retroperitoneal space [4].

The anesthetic–surgical stress response encompasses metabolic, neuroendocrine, hemodynamic, immunological, and behavioral components [6,7,8]. However, it is also well known that sustained, constant, and uncontrolled states cause unwanted effects during the postoperative recovery phase [5]. While the inflammatory response is a crucial defense mechanism against infection and initiates tissue repair, a prolonged or uncontrolled response can lead to adverse effects such as excessive pain, immunosuppression, organ dysfunction, and even death [6].

Oxidative alterations, such as plasma protein thiols (P-SHs) and plasma and erythrocyte thiobarbituric acid reactive substances (TBARSs), after OVH [5,9,10] and ovariectomy [10] have been reported in dogs. These alterations result in an imbalance favoring oxidants over antioxidants, leading to cellular damage and potentially contributing to various diseases [6]. A comprehensive profile using various integrated biomarkers of total antioxidant status, individual antioxidants, and oxidative stress indicators, such as lipid peroxidation and reactive oxygen species (ROS) production, can provide valuable insights into oxidative stress in dogs [11,12].

Accordingly, the choice of intraoperative approaches, techniques, and materials often focuses on minimizing trauma [5,7] and reducing the operative time [2,5]. This study aimed to compare the surgical stress response in healthy bitches undergoing open OVH, with or without OSL rupture.

## 2. Materials and Methods

### 2.1. Animals

Animal handling and procedures were approved by the faculty’s Animal Experimentation Ethics Committee at the Universidade Federal da Fronteira Sul, Brazil (Protocol No. 676,103,022 CEUA). The study included 20 healthy female dogs of various breeds with a body condition score between 5 and 6 (Scale 1 to 9). Prior to inclusion, each dog underwent a detailed physical examination, complete white blood cell count (WBC), serum alkaline phosphatase and alanine aminotransferase activity analysis, serum creatinine and total plasma protein level analysis, abdominal ultrasound, and electrocardiogram. Significant macroscopic changes in the uterus, ovaries, and uterine tubes were evaluated post-surgery as the exclusion criteria. Animals that were pregnant, lactating, or in a heat cycle were not included in the study.

Patients were randomly allocated into two groups based on the surgical technique: OVH with OSL rupture (OSL-R, *n* = 10) and OVH without rupture (OSL-NR, *n* = 10).

### 2.2. Surgical Preparation and Anesthesia

The patients were housed in boxes at 23 °C for 48 h prior to surgery, allowing them to acclimate to the researchers and other animals. They remained in the same room for 48 h post-surgery for continuous postoperative assessments. The animals received food and water ad libitum and had access to a restricted outdoor area for defecation and urination. All procedures were performed by the same team to maintain consistency in the methodology.

After an eight-hour food and four-hour water fast, the dogs received methadone (0.4 mg/kg IM) (Mytedon, Cristalia, São Paulo, Brazil). Fifteen minutes later, a broad abdominal trichotomy was performed, and intravenous access was established with Ringer’s Lactate solution (5 mL/kg/h IV), which was administered until extubation. Anesthesia was induced with propofol (1 mg/kg IV every 30 s) (Fresofol, Fresenius-Kabi, São Paulo, Brazil) until the loss of the laryngotracheal reflex, allowing for tracheal intubation and maintenance with 100% oxygen and isoflurane via a calibrated vaporizer. Fentanyl citrate (2.5 µg/kg IV) (Janssen-Cilag Farmacêutica LTDA, São Paulo, Brazil) was administered if a 20% increase in any two baseline parameters (cardiac and respiratory rates, systolic blood pressure, or temperature) was detected; these animals were excluded from statistical analysis following medication administration. Immediately before (Basal) and during the surgery, physical parameters—including the cardiac and respiratory rates, systolic blood pressure, and esophageal temperature—were recorded during intubation (M1), during right (M2) and left (M3) ovarian vessel manipulation, and at extubation (M4).

At the end of the surgery, while the patient was still under anesthesia, a 16G catheter was inserted into the jugular vein, secured with sutures and instant adhesive, and covered with a sterile bandage to facilitate blood sampling during the postoperative period.

### 2.3. Surgical Procedures

All the procedures were performed by the same team, composed of a qualified and experienced surgeon, an anesthesiologist, a surgical assistant, and a circulating nurse. The patients were positioned in dorsal recumbency and draped with a single layer of sterile, non-reusable drape material. Both groups underwent a retro-umbilical celiotomy, comprising one-third of the distance between the umbilicus and pubis, following a standardized surgical protocol. In the OSL-R group, manual rupture of the ligament was performed, while in the OSL-NR group, the ligament was left intact. Hemostasis of the ovarian and uterine blood vessels was achieved using a 5 mm permanent laparoscopic bipolar forceps (EDLO, Porto Alegre, Brazil). Muscular and subcutaneous closure was completed using a buried continuous subcutaneous suture with polyglactin 910 (Shalon, Goiás, Brazil), followed by an intradermal suture with nylon. Immediately after surgery, the patients received methadone (0.3 mg/kg IM, TID for 48 h), daily wound care with chlorhexidine (every 24 h), and protective clothing. The time of the procedures was recorded. Following surgery, the surgeon and the surgical assistant evaluated the difficulty of ovarian vessel coagulation, scoring it as 0, +, ++, +++, or ++++. The “0” was used if there was no difficulty, “+” for minor difficulty, “++” for medium difficulty, “+++” for moderate difficulty, and “++++” for major difficulty.

### 2.4. Postoperative, Physical, and Pain Evaluations and Blood Sampling

The procedures were conducted by the same researchers immediately before surgery and then at 2, 6, 12, 24, and 48 h, as well as at 10 days post-surgery.

The physical examination included assessments of the rectal temperature, respiratory rate, heart rate, and systolic blood pressure (via Doppler) at the specified time points. Pain evaluation was conducted by two experienced veterinarians blinded to the surgical technique, using the Visual Analog Scale (VAS) [13] and the short-form Glasgow Composite Measure Pain Scale (CMPS-SF) [14]. Analgesic rescue with morphine (0.3 mg/kg IM) was administered if the patients reached scores of 3/10 on the VAS or 6/24 on the CMPS-SF. Evaluations were conducted every 2 h up to 12 h post-surgery and every 4 h up to 24 h post-surgery. Patients receiving analgesic rescue were excluded from further statistical analysis following medication administration.

For laboratory tests, 7 mL of blood was collected at the specified time points via a jugular catheter or venipuncture, divided among three tubes, and processed. The EDTA tube (4 mL) was used for the hematocrit, total plasma protein, leukogram, oxidative stress aliquots, and erythrocyte sedimentation rate; the sodium heparin tube (2 mL) was used for the NBT test; and the fluoride tube (1 mL) (Glistab, Labtest, São Paulo, Brazil) was used for glucose analysis.

### 2.5. Laboratory Analysis

Hematocrit levels were determined using the microhematocrit technique [15]. Total plasma protein was measured with a refractometer, using plasma from the capillary tube. White blood cell counts were manually assessed, with differential leukocyte counts performed in duplicate on slides stained using the Diff-Quick method and analyzed under an optical microscope at 1000× magnification (Olympus^®^ CX21, Tokyo, Japan). The leukocyte assessment included segmented neutrophils, lymphocytes, monocytes, and eosinophils [16], followed by data evaluation [17].

The Westergren method to evaluate the erythrocyte sedimentation rate was measured using 1 mL of EDTA blood mixed with 250 µL of 0.9% NaCl. The samples were placed in a reservoir tube, and the sedimentation rate was recorded in millimeters after one hour [18]. For the NBT test, 2 mL blood was placed in a heparinized tube, and nitroblue tetrazolium dye (NBT, Amresco^®^, Solon, OH, USA) was added to assess neutrophil oxidative metabolism. Slides were prepared in duplicate and stained using the Diff-Quick method, with 100 neutrophils per slide evaluated at 1000× magnification by the same researcher. For the stimulated NBT test, Saccharomyces cerevisiae (Zymosan^®^, Sigma-Aldrich, St. Louis, MO, USA) was added, and duplicate slides were prepared, stained with the Diff-Quick method, and examined under an optical microscope (Olympus^®^ CX21, Tokyo Japan), with 100 neutrophils assessed per slide at 1000× magnification [19].

Glucose samples were stored at −80 °C and thawed simultaneously for analysis, which was performed using an automatic analyzer (CM250, WienerLab^®^), followed by data interpretation [17].

For oxidative metabolism evaluation, blood samples were placed in serum separator tubes and centrifuged at 3000× *g* for 10 min. Each sample was then aliquoted (400 µL per tube) and stored at −80 °C until analysis. Immediately before evaluation, the samples were thawed. The biomarkers included vitamin C [20,21], plasma protein thiols (P-SHs) [22], erythrocyte reduced glutathione (GSH) [22], and plasma and erythrocyte thiobarbituric acid reactive substances (TBARSs) [23,24].

### 2.6. Statistical Analysis

Data normality was assessed using the Shapiro–Wilk test, and the homogeneity of variances was evaluated with Levene’s test. For comparisons between groups with normally distributed data and homogeneous variances, an uncorrected t-test was applied. When variances were heterogeneous, Welch’s correction was used. For non-normally distributed data, a one-way ANOVA was performed, with Tukey’s post hoc test applied for homogeneous variances and Games–Howell’s test for heterogeneous variances. For comparisons of time points within the same group, normally distributed data were analyzed using one-way ANOVA with Tukey’s post hoc test. For non-normally distributed data, the Kruskal–Wallis test was applied, followed by the Dwass–Steel–Critchlow–Fligner post hoc test. All statistical analyses were performed using Jamovi software version 2.3 (2022) [25].

## 3. Results

The animals included in this study, selected for elective OVH at a Veterinary Teaching Hospital, represented various breeds. Of these, eight had a deep chest conformation, with five assigned to the OSL-R group and three to the OSL-NR group. Two patients had prior pregnancies, with one allocated to each group. The body condition scores were similar between groups (*p* = 1.0, 5.3 ± 0.48). The patients’ weights ranged from 10 to 23.4 kg; the OSL-R group had an average weight of 18.14 ± 3.68 kg, while the OSL-NR group had an average of 15.71 ± 4.32 kg (*p* = 0.19). The OSL-R animals had an average age of 25.2 ± 7.31 months, and the OSL-NR animals averaged 21.5 ± 14.59 months (*p* = 0.48). According to the surgeon and assistant’s assessment, the difficulty in exposing the ovarian arteriovenous complex was similar between the groups (*p* = 0.93).

The surgeries and anesthetic procedures proceeded without complications. The surgical times were similar between groups (OSL-R = 17 ± 2.35 min, OSL-NR = 15.8 ± 2.04 min; *p* = 0.24). All wounds healed adequately, and the skin sutures were removed 10 days post-surgery. At this time, one animal in the OSL-R group experienced gastrointestinal upset and vomiting; only the data (10 days) from this patient were excluded from the study. Another patient in the same group developed local infection and was treated with cephalexin (30 mg/kg TID for 8 days); due the absence of systemic alteration, these data were maintained in the study. 

There was no difference between groups regarding the first water and solid intake (*p* = 0.24, OSL-R = 214 ± 178 min and OSL-NR = 325 ± 190 min; *p* = 0.65, OSL-R = 255 ± 198 min and OSL-NR = 292 ± 1.52 min, respectively). The time to first urination (*p* = 0.06; OSL-R = 126 ± 79.2 min, OSL-NR = 251 ± 179 min) and first defecation (*p* = 0.18; OSL-R = 263 ± 295 min, OSL-NR = 500 ± 417 min) were similar between groups.

During the surgery, four animals received analgesic rescue, with three being from OSL-NR and one from OSL-R, all at M3. The data from this patient were removed after the analgesic application. Regarding transoperative physical parameters, no differences were observed between groups or over time for systolic blood pressure (*p* > 0.05). The heart rate in the OSL-R group decreased at M1, M2, and M3, with a further reduction at M4 (*p* = 0.05). The respiratory rate showed differences over time in both groups; in the OSL-R group, the respiratory rate decreased at M1 and again at M2, M3, and M4 (*p* = 0.003); in the OSL-NR group, the respiratory rate decreased at M1 and remained stable at M2, M3, and M4 (*p* < 0.001). The rectal temperature in the OSL-NR group decreased at M1 and further decreased at M2, M3, and M4 (*p* = 0.01).

During the postoperative period, two animals of the OSL-R group received analgesic rescue, with one being at T6 and the other at T8; one animal of the OSL-NR group received rescue at T4. The data of these patients were excluded from the statistical analysis after this time. The VAS and CMPS-SF showed no significant differences between groups, although both groups presented changes over time. In the VAS, the OSL-R group showed an increase in scores from 2 h until 24 h, followed by a decrease at 48 h, while the OSL-NR group exhibited an increase in scores from 2 h until 48 h. In the CMPS-SF, the OSL-R group reached peak scores at 2 h and 4 h, returning to baseline after 6 h (*p* < 0.001). The OSL-NR group peaked at 2 h, followed by score reductions from 4 h to 24 h and another decrease at 48 h (*p* < 0.001).

No significant differences were observed between groups regarding the systolic blood pressure, respiratory rate, or rectal temperature. Over time, the rectal temperature in the OSL-NR group decreased at 6 h and returned to baseline after 10 days. The heart rate differed between groups, with the OSL-R group showing higher values at 12 h (*p* = 0.02; OSL-R = 97.2 ± 17.2, OSL-NR = 80.8 ± 10). The heart rate also varied over time in both groups: in the OSL-R group (*p* < 0.001), the rates were elevated at 10 days, while in the OSL-NR group (*p* < 0.001), the values decreased from 6 h to 48 h and were higher than baseline at 10 days. Only the OSL-R group exhibited changes in the respiratory rate over time (*p* < 0.001), with higher values observed at 10 days. The rectal temperature in the OSL-R group also presented elevated values at 10 days (*p* < 0.001).

Plasmatic total protein and hematocrit levels were monitored to detect any signs of postoperative bleeding or dehydration in individual patients; however, no such events were observed at any evaluation point. Additionally, there were no significant differences in the glucose levels or erythrocyte sedimentation rates between groups or over time.

For total leukocyte levels (Figure 1), differences were noted between groups at 6 h, with higher counts in the OSL-R group (*p* < 0.008). Over time, both groups showed variations: in the OSL-R group, the counts peaked at 6 h and 12 h, decreased at 24 h, and returned to baseline at 48 h post-surgery. In the OSL-NR group, the counts peaked at 6 h, decreased at 12 h, and normalized by day 10.

Segmented neutrophils showed a significant difference between groups at 6 h (Figure 1), with higher values in the OSL-R group (*p* = 0.004; OSL-R = 17,037 ± 3056 and OSL-NR = 12,984 ± 1992). Over time, both groups exhibited significant changes (*p* < 0.001): in the OSL-R group, the counts peaked at 6 h, decreased by 12 h, and returned to baseline at 48 h, while in the OSL-NR group, peaks were observed at 6 h, 12 h, and 24 h, followed by a reduction at 48 h and a return to baseline by day 10.

Lymphocyte measurements remained stable, with no differences between groups or over time (*p* > 0.05). Eosinophil counts differed between groups at 48 h (Figure 1), with lower values observed in the OSL-R group (*p* = 0.025; OSL-R = 432 ± 398, OSL-NR = 1101 ± 703), although no significant temporal changes were observed within either group.

Monocyte counts showed differences between groups at 6 h (*p* = 0.03; OSL-R = 926 ± 529, OSL-NR = 1520 ± 601), 24 h (*p* = 0.01; OSL-R = 1087 ± 517, OSL-NR = 1794 ± 481), and 48 h (*p* = 0.03; OSL-R = 878 ± 445, OSL-NR = 1507 ± 658) (Figure 1). Over time, no significant variations were observed within the OSL-R group (*p* = 0.09). However, in the OSL-NR group (*p* < 0.001), the monocyte counts rose at 6 h, declined at 12 h, peaked again at 24 h, and returned to baseline at 48 h.

For the unstimulated NBT test, significant differences were observed between groups at 6 h (*p* = 0.023; OSL-R = 10.1 ± 7.69, OSL-NR = 3.25 ± 2.82) and at 24 h (*p* = 0.038; OSL-R = 12.67 ± 5.52, OSL-NR = 20.78 ± 8.98) (Table 1). The OSL-NR group showed temporal variation (*p* < 0.001), with neutrophil oxidative metabolism increasing at 12 h, peaking at 24 h, decreasing by 48 h, and returning to baseline by day 10. No significant differences were found between groups or over time for the stimulated test assessing phagocytic function (Table 1).

Regarding oxidative metabolism, the vitamin C levels differed between groups at 6 h (*p* = 0.02; OSL-R = 114 ± 35.8, OSL-NR = 70.1 ± 33.7) and at 12 h (*p* = 0.01; OSL-R = 100 ± 52.2, OSL-NR = 40.3 ± 35.4) (Figure 2). No temporal variations were observed within either group. Both the GSH and P-SH levels showed no significant differences between groups or over time (Figure 2). The plasma TBARS levels differed between groups at 6 h, being lower in the OSL-NR group (*p* = 0.03; OSL-R = 60.8 ± 43.0, OSL-NR = 26.6 ± 11.9) (Figure 2), with the OSL-NR group showing temporal variation (*p* = 0.04), peaking at 24 h and returning to baseline by 48 h. Erythrocyte TBARSs also differed between groups at 12 h (*p* = 0.01; OSL-R = 63.4 ± 22.7, OSL-NR = 38.8 ± 14.4) and at 24 h (*p* = 0.03; OSL-R = 67.2 ± 28, OSL-NR = 42.2 ± 18.5) (Figure 2). In the OSL-R group, erythrocyte TBARSs increased at 6 h and returned to baseline by 12 h (*p* = 0.007) (Figure 2).

## 4. Discussion

Stress is an adaptive response essential for a favorable reaction to harmful stimuli, engaging metabolic, endocrine, hemodynamic, behavioral, and immunological mechanisms [6]. Thus, this study was designed to investigate a frequent maneuver used during OVH [1,2,3]. The digital rupture of the OSL is a standard technique when exposure of the ovarian vessels is required for safe hemostasis [26]. It alleviates tension, bringing the ovarian vessels into view and facilitating effective hemostasis.

In pregnant dogs or those with an enlarged uterus, and depending on the extent of the abdominal approach, it is possible to expose the ovarian vessels adequately for ligation without OSL rupture [3]. Two animals in this study had been pregnant once before, and the surgical time and difficulties observed for these animals did not differ from the others.

Ovarian vessel cauterization was performed in three deep-chested animals without OSL rupture. While this rupture is generally required to allow for the retraction of the ovary, especially in deep-chested dogs and with smaller abdominal incisions [3], it did not pose a limiting factor or increase the difficulty of the procedure, as assessed by both the surgeon and the assistant (*p* = 0.93). Furthermore, the bipolar forceps facilitated and promoted efficient, effective hemostasis [2,5].

Despite age and weight standardization efforts, some procedures took longer, particularly in larger animals, as described [1,26]. It is important to note that the body condition score could potentially serve as a confounding factor when assessing the overall surgical time. Dogs with higher body condition scores often experience longer and more challenging surgeries, partly due to the increased fat around their ovaries and OSL [26], such as dogs more than 25 kg [1].

In the present investigation, one case of gastrointestinal upset and vomiting and one of wound infection were reported in the OSL-R group (10%). Complications associated with OVH typically range from 7.5% to 19%, with the most common being minor issues, such as incision site inflammation and gastrointestinal upset. Other reports cite inflammation or infection (5.6%), hemorrhage (2.8%), and pancreatitis (0.7%) [1]. Infections are generally linked to immune compromise post-surgery, as tissue manipulation increases tissue susceptibility to bacterial growth [27]. Although no specific cause was identified for this case, it is recognized that pain and surgical stress can suppress the immune system and predispose to infection [6].

It is essential to recognize that there is no single, definitive biomarker capable of evaluating pain or stress in animals. Research on surgical and anesthetic stress must involve physiological indicators (heart and respiratory rates, blood pressure, and temperature), biochemical markers (cortisol, catecholamines, lactate, glucose, and IL), behavior-based pain scoring systems, and assessments of autonomic responses to noxious stimuli [6]. This study evaluated various surgical and physical parameters and laboratory tests to improve the prevention and management of neurobiological and behavioral responses throughout the perioperative period [6,28].

The transoperative physical parameters did not differ between groups (*p* > 0.05), contrary to what was published in a study using a different anesthetic protocol. Nociception triggers physiological responses, such as an increased heart rate, respiratory rate, and blood pressure, which are typically observed in bitches during the manipulation of the ovaries and OSL [26]. In this study, although one animal in the OSL-R group required analgesic rescue, three animals in the OSL-NR group also required it. This finding suggests that OVH without OSL rupture may be more painful, possibly due to continuous ovarian manipulation during bipolar coagulation. In addition, we believe that three-clamp modified techniques may, in fact, be more traumatic than bipolar coagulation, potentially exacerbating the subtle differences observed in this study. This study found that the temperature, heart rate, and respiratory rate significantly decreased after surgery but returned to baseline levels at recall. These parameters can be influenced by multiple factors, such as premedication and anesthesia [29].

Although no differences were observed in the VAS and CMPS-SF between groups in this study, two animals in the OSL-R group and one in the OSL-NR group required analgesic rescue, which contrasts with the transoperative findings. This aligns with previous studies, which suggest that OSL manipulation and rupture during open OVH is more painful than cauterization with no rupture [4,30].

Over time, the VAS showed differences only from baseline to the other time points (*p* < 0.001). This scale was used combined with the CMPS-SF to address its subjective nature, allowing the evaluators to distinguish between pain and behavioral changes [31]. Unlike previous observations, in this study, the patients exhibited a respiratory rate, heart rate, and rectal temperature at or above baseline levels at recall, indicating similar psychological stress in these moments [29].

Regarding the CMPS-SF, the pain scores returned to baseline levels at 6 h (*p* < 0.001), suggesting that acute OSL rupture causes a short-term pain effect. In contrast, the OSL-NR group experienced a greater pain stimulus at 2 h, which diminished from 4 h to 24 h (*p* < 0.001), indicating a more prolonged pain response compared to OSL-R. Previous studies have shown that open OVH with OSL rupture results in higher pain stimuli from 2 h to 12 h [4,32,33]. However, the results of this study indicated a milder but more prolonged pain stimulus lasting up to 24 h (*p* < 0.001). When sealing and dividing the ligament with the vessel sealing device, minimal tension was applied to the ligament. Nonetheless, it is possible that despite the lack of tension, division of the OSL with the vessel sealing device caused more stimulation than anticipated [30].

A 48 h acclimation period was implemented in this study to help the veterinarians distinguish between behaviors related to fear, agitation, or aggression and those associated with pain. This is essential, as the central nervous system’s response to potentially harmful stimuli triggers behavioral and autonomic mechanisms aimed at restoring homeostasis [6,29], which may influence clinical and laboratory outcomes. Previous research underscores that confinement in an unfamiliar, uncontrolled environment—combined with handling by unfamiliar individuals—represents a substantial stressor detectable as early as 30 min after caging [28].

No significant differences were observed in the glucose levels (*p* > 0.05). Although cortisol was not evaluated in this study, it typically increases during the perioperative period, and consequently, alters the glucose level [6]. Contrary, other study identified glucose level alterations [8]. This observation suggested a smaller elevation of the cortisol level, which predisposes to multiple postoperative complications [32]. In this same sense, there was no difference between groups regarding the first water and solid intake, nor the first urination and defecation. In contrast, the mean of solid ingestion of other studies was 2 h after the procedure, which was earlier than this study [30]. Following surgery, the immune system may either be activated or suppressed depending on the type of stress response induced [6]. In the OSL-R group, elevated leukocyte (*p* < 0.001) and segmented neutrophil counts (*p* < 0.001), along with eosinopenia, suggest a more intense inflammatory reaction compared to the OSL-NR group, which did not present eosinopenia. Over time, the sustained increase in segmented neutrophils and monocytosis, persisting up to 10 days, suggests a prolonged inflammatory response following OSL-NR, consistent with prior findings; it showed extended neutrophilia and monocytosis but transient lymphopenia and eosinopenia limited to the early postoperative period [9,28]. A possible explanation is that glucocorticoids may contribute to a stress leukogram [6]. In the current study, the OSL-R group displayed these alterations, with the exception of lymphopenia.

In veterinary medicine, the erythrocyte sedimentation rate (ESR) has been used as an indicator in dogs with leishmaniasis, dirofilariasis, ehrlichiosis, and osteoarthritis [34,35,36,37]. Although positive ESR correlations have been reported, this study found no differences between groups or across time points (*p* > 0.05). It could be attributed to the acute nature of the inflammatory response in these patients, which contrasts with the chronic inflammation described [34].

Surgery induces inflammation and oxidative stress, which can have a significant impact on postoperative complications [32]. Reactive oxygen species (ROS) are produced as part of the inflammatory response and can initiate a chain reaction. Neutrophils also generate ROS in a process called respiratory or oxidative burst, an antimicrobial and inflammatory mechanism [7,38]. However, the suppression of the neutrophil oxidative metabolism can suppress immune responses, increasing the risk of infections [39].

Inflammatory processes and infections have been shown to enhance neutrophil oxidative metabolism [40,41]. In the present study, inflammation likely contributed to the increased oxidative metabolism observed at 24 h (*p* = 0.03), especially in the OSL-NR group (*p* < 0.001). A correlation between the increase of the neutrophil oxidative metabolism and increase in erythrocyte and plasma TBARSs at 24 h should be pointed. Another study observed no correlation between NBT and TBARSs and suggested that neutrophil oxidative metabolism increases during pyometra did not cause lipid peroxidation [40].

Vitamin C showed a difference between groups at 6 h (*p* = 0.02) and 12 h (*p* = 0.01) post-surgery, with superior measurements in the OLS–R group, which may be a compensatory response, as observed in dogs with congestive heart failure [42] Vitamin C is a ubiquitous antioxidant that plays a crucial role in neutralizing various reactive oxygen species (ROS), which can regenerate other oxidized antioxidants [43].

The P-SH evaluation, in this study, does not present differences between groups (*p* > 0.05), nor along the time, opposite to what has been described, when this plasmatic protein antioxidant decreases its concentration after surgery compared with the preoperative level [12]. The result of this study can suggest a depletion of the antioxidant defense mechanism postoperatively [12]. 

The thiol group is a potent reducing agent, making GSH the most prevalent intracellular antioxidant molecule, being crucial for cellular defense [44]. In the present study, the GSH levels did not show significant differences between groups or over time (*p* > 0.05), consistent with previous findings [9,45]. A study has shown that its levels significantly decrease at one and six months post-OVH in dogs, suggesting a delayed recovery phase in GSH levels following surgery [46]. 

Prior studies indicate that dogs undergoing elective OVH with OSL rupture exhibit elevated TBARS levels after 24 h post-surgery, reflecting increased lipid peroxidation [9]. Similarly, other studies reported increased malondialdehyde (MDA) 24 h after OVH, a breakdown product of TBARSs [47]. Corroborating the findings here, the plasmatic TBARSs at 6 h (*p* = 0.03) and erythrocyte TBARSs at 12 h (*p* = 0.01) and 24 h (*p* = 0.03) were higher after OSL-R than OSL-NR; these results are similar to those mentioned. A comparable pattern was seen with open OVH, showing a TBARSs increase up to 72 h post-surgery [10]. Another investigation also found that ovariectomy elevated plasmatic TBARS levels up to 30 days afterward, suggesting long-term oxidative stress [12].

The statistical power of this study was limited due to the restricted sample available, which was selected based on their availability and specific clinical conditions. Consequently, we included animals of varying breeds, sizes, and body conformation, introducing variability within each group. While this heterogeneity reflects the diversity typically encountered in clinical practice and enhances the external validity of the findings, it likely reduced the ability to detect significant differences. In this sense, therefore, the data were thoroughly analyzed and discussed. In this sense, this study underscores the importance of thorough expertise by both veterinary surgeons and anesthesiologists in managing the multifaceted aspects of perioperative stress and its consequences to control or, ideally, prevent stress responses [6]. The manipulation of the ovarian suspensory ligament is a common maneuver intended to aid in hemostasis during surgery [1,3]. However, it may contribute to complications, particularly when coupled with other comorbidities, potentially leading to severe outcomes such as organ failure or, in extreme cases, death [6].

## 5. Conclusions

The rupture of the ovarian suspensory ligament (OSL) in healthy female canines suggests a more pronounced response to surgical stress, considering the total white blood count, eosinophil and monocytes counts, and the plasmatic and erythrocyte thiobarbituric acid reactive substances measurements. It is suggested that OSL rupture may trigger a more intense nociceptive stimulus during surgery, whereas OSL preservation tends to elicit more pain in the postoperative period.

## Figures and Tables

**Figure 1 vetsci-11-00658-f001:**
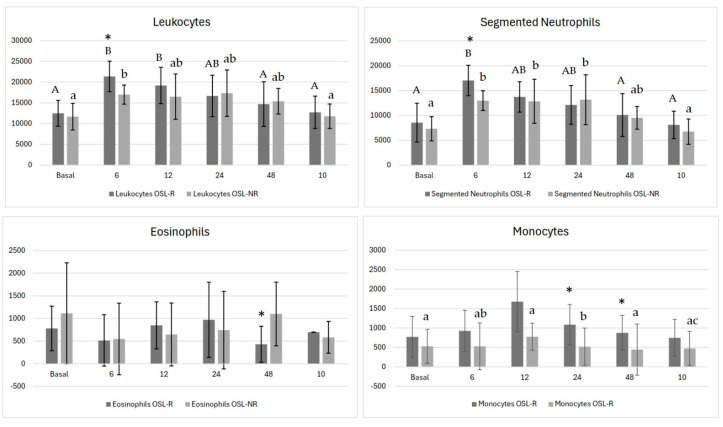
Mean values of total leukocytes, segmented neutrophils, eosinophils, and monocytes in bitches undergoing ovariohysterectomy via celiotomy with ovarian suspensory ligament rupture (OSL-R) and without rupture (OSL-NR). Asterisks (*) indicate statistically significant differences between groups (*p* < 0.05). Uppercase different letters represent differences across time points within the OSL-R group, while lowercase different letters represent differences across time points within the OSL-NR group.

**Figure 2 vetsci-11-00658-f002:**
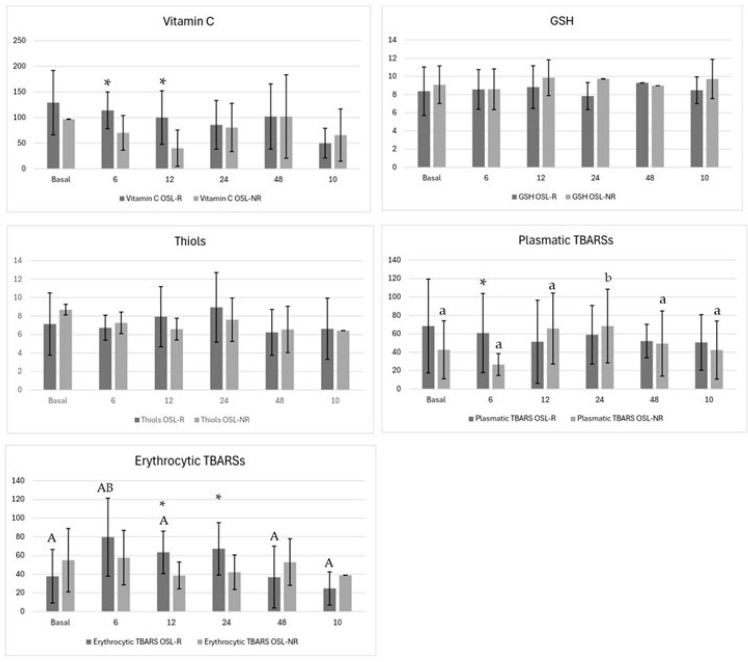
Means and standard deviations of the vitamin C, plasma protein thiols (P-SHs), erythrocyte reduced glutathione (GSH), plasma and erythrocyte thiobarbituric acid reactive substances (TBARSs) of bitches that underwent ovariohysterectomy after ovarian suspensory ligament rupture (OSL-R) or not (OSL-NR). Asterisks (*) indicate statistically significant differences between groups (*p* < 0.05). Uppercase different letters represent differences across time points within the OSL-R group, while lowercase different letters represent differences across time points within the OSL-NR group.

**Table 1 vetsci-11-00658-t001:** Means and standard deviations of the unstimulated and stimulated NBT test of bitches that underwent ovariohysterectomy after ovarian suspensory ligament rupture (OSL-R) or not (OSL-NR). The columns represent surgery techniques and the lines the evaluation times. Lowercase letters denote differences across time points within the OSL-NR group.

NBT Unstimulated Test			
Times	OSL-R	OSL-NR	*p* Value
Basal	10.2 ± 7.74	10.20 ± 4.85a	0.99
6	10.1 ± 7.69	3.25 ± 2.82a	0.02 *
12	5.56 ± 6.52	9.30 ± 8.96ac	0.31
24	12.67 ± 5.52	20.78 ± 8.98b	0.03 *
48	9.10 ± 6.54	15.11 ± 9.55abc	0.13
10	10.22 ± 7.24	7.24 ± 7.88ac	0.71
*p* value	*p* = 0.26	*p* < 0.001 *	
NBT Stimulated Test			
Times	OSL-R	OSL-NR	*p* Value
Basal	8.8 ± 62.8	9.60 ± 73.9	0.71
6	8.56 ± 42.9	6 ± 33.7	0.34
12	8.30 ± 52.2	8.33 ± 35.4	0.99
24	8.20 ± 47.6	10.8 ± 47.1	0.32
48	9.20 ± 63.6	8.20 ± 81.4	0.68
10	7.00 ± 29.0	10.8 ± 51	0.07
*p* value	*p* = 0.98	*p* = 0.18	

Asterisks (*) indicate statistically significant differences between groups (*p* < 0.05).

## Data Availability

All data is contained within this paper.

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
