# Peer review of "Impact of Ovarian Suspensory Ligament Rupture on Surgical Stress in Elective Ovariohysterectomy in Bitches"

_vetsci, 2024, doi:10.3390/vetsci11120658_

Round 1
Reviewer 1 Report
Comments and Suggestions for Authors
“Impact of Ovarian Suspensory Ligament Rupture on Surgical 2 Stress in Elective Ovariohysterectomy in Bitches” is an essential observation in veterinary surgery.
My comments are as follows –
The introduction, methods, and results flow well. However, when I went to the discussion, I needed to catch up at different times. It seems the authors discussed quite a lot. I would appreciate discussing only their results with other observations on the specific findings.
In the whole study, I found (OSL-R, n = 10) and OVH without rupture (OSL-NR, n = 10). However, later it was explained animals having different issues, were excluded from the analysis. Therefore, the author should put the n=? in every case, wherever presenting results in the graph or text. I am also concerned about the statistical power of the analysis. The authors should mention the statistical power. The authors should discuss the study's limitations at a certain point in the discussion.
Were all the operations performed by the same surgeon or a group of the same or different surgeons?
Line 133-135: “Following surgery, the surgeon and an assistant 133 evaluated the difficulty of ovarian vessel coagulation, scoring it from 0 to +, ++, +++, or 134 ++++, and recorded the time taken for the procedure.” can the author please explain a bit here, what does 0 to ++++ means. This will help the reader to understand correctly. Additionally, the authors need to add a reference here for such techniques.
Authors could add mean values rather than median values in the table.
Author Response
We would like to thank the reviewer for the compliment and all the constructive consideration. We modified the text according to the suggestions.
1- The introduction, methods, and results flow well. However, when I went to the discussion, I needed to catch up at different times. It seems the authors discussed quite a lot. I would appreciate discussing only their results with other observations on the specific findings. Answer: We evaluated the discussion and removed the redundant information and consolidated data, as suggested. The alterations were highlighted.
2- In the whole study, I found (OSL-R, n = 10) and OVH without rupture (OSL-NR, n = 10). However, later it was explained animals having different issues were excluded from the analysis. Therefore, the author should put the n=? in every case, wherever presenting results in the graph or text.
Answer: the results were clarified in the text;
- “Another patient in the same group developed local infection and was treated with cephalexin (30 mg/kg TID for 8 days); due the absence of systemic alteration, this data was maintained in the study.” (Line 222 - 224).
- During the surgery, four animals received analgesic rescue, being three of OSL-NR and one OSL-R, all at M3. The data of this patient was removed after the analgesic application (Line 230 - 232).
- At postoperative, two animals of the OSL-R received analgesic rescue, being one at T6 and other at T8; one animal of the OSL-NR was rescued at T4. The data of these patients were excluded from the statistical analysis after this time (Line 241-243).
3- I am also concerned about the statistical power of the analysis. The authors should mention the statistical power. The authors should discuss the study's limitations at a certain point in the discussion.
Answer: The statistical power of this study was limited (0.18) due to the restricted sample available. This limitation arose because we use routine patients, who were selected based on their availability and specific clinical conditions. Consequently, we included animals of varying breeds, sizes and body conformation, introducing variability within each group. While this heterogeneity reflects the diversity typically encountered in clinical practice and enhances the external validity of the findings, it likely reduced the ability to detect statistically significant differences (Line 521-533).
4- Were all the operations performed by the same surgeon or a group of the same or different surgeons?
Answer: The text was modified to clarify the doubt (Line 127 - 128).
5- Line 133-135: “Following surgery, the surgeon and an assistant 133 evaluated the difficulty of ovarian vessel coagulation, scoring it from 0 to +, ++, +++, or 134 ++++, and recorded the time taken for the procedure.” can the author please explain a bit here, what does 0 to ++++ means. This will help the reader to understand correctly. Additionally, the authors need to add a reference here for such techniques.
Answer: We altered this paragraph to improve the understanding (Line 141-144). In response to the scale reference, the authors are unaware of one scale which could be used for this evaluation; in this way, we create this scale.
6- Authors could add mean values rather than median values in the table.
Answer: “Median” was substituted for “Mean” in the title of Table 1 and Figure 2.
Reviewer 2 Report
Comments and Suggestions for Authors
The work is interesting. With the growth of laparoscopic surgery the resection of the ligament of the ovary will be performed less and less and that in any case in open surgery it is important to allow correct exposure of the ovary.
Despite this, it is important to know if this technique creates important alterations to the bitch and the evaluation of oxidative metabolism is very interesting.
Precisely because multiple postoperative complications often are associated with elevated cortisol and it is the most important indicator of stress, about me it’s very important that this hormone is evaluated.
It should be made clearer whether cortisol has been evaluated, what results have been obtained and, if so, include cases in which the evaluation has taken place
Line 406: It is not clear whether cortisol was assessed or not in this study
Line 409: where does it say that cortisol has been measured and no differences are found?
Author Response
The work is interesting. With the growth of laparoscopic surgery the resection of the ligament of the ovary will be performed less and less and that in any case in open surgery it is important to allow correct exposure of the ovary.
Despite this, it is important to know if this technique creates important alterations to the bitch and the evaluation of oxidative metabolism is very interesting. Precisely because multiple postoperative complications often are associated with elevated cortisol and it is the most important indicator of stress, about me it’s very important that this hormone is evaluated. It should be made clearer whether cortisol has been evaluated, what results have been obtained and, if so, include cases in which the evaluation has taken place.
Line 406: It is not clear whether cortisol was assessed or not in this study
Line 409: where does it say that cortisol has been measured and no differences are found?
Answer: We appreciate the reviewer’s comments regarding this matter. We hope the article will be of great value to the readers and we modified the text according to your consideration.
The methods, results and conclusion were improved as suggested. Unfortunately, cortisol measurement was not feasible due to budget constraints; although, we believe it would have been valuable in interpreting the results of this study. To ensure the robustness of the research, we evaluated physiological parameters (heart rate, respiratory rate, blood pressure, temperature, appetite, urination, etc.), laboratory tests and markers (leukogram, and oxidative metabolism), as indirects signs of all stress hormones (cortisol and catecholamines). We addressed this consideration in the discussion (Lines 422-433).
Regrettably, the research design included a limited sample size 'n' and comprised patients with significant variability (e.g., differences in breed, size, and body conformation). This study was conducted using the available routine patients, which can introduce bias into the results. Therefore, the data were thoroughly analyzed and discussed (Line 521-533).
Reviewer 3 Report
Comments and Suggestions for Authors
This article has some advantages in terms of introduction, study design, and presentation of results, but there are also some shortcomings. The introduction provides the necessary background and some references, but it can be more in-depth and updated. The study design was reasonable and the methods were described in detail, however some details need to be improved. The results are clear and support the conclusions well, but the complexity of the results needs to be better explained in the discussion. Overall, the articles have some academic value, but improvements in some aspects are needed to improve the quality and impact of the research.
The current sample size on the findings should be discussed, such as whether the sample size may have prevented some effects from being detected or increased the risk of false negative results because the sample size was too small. Consider whether a sensitivity analysis of sample size is needed to assess the effect of variation in sample size on the stability of the study conclusions.
For some complex results, such as cases where the trend of some indicators at different time points is inconsistent or does not correspond to expectations, preliminary explanations or speculations should be provided in the results section. For example, when discussing the changes in vitamin C levels, combining the surgical stress response and oxidative stress mechanisms, explaining why the OSL-R group had higher vitamin C levels in the early postoperative period and the possible association of this change with other relevant indicators (such as oxidative stress indicators, inflammatory response indicators, etc.) helps readers better understand the significance of the results.
Author Response
This article has some advantages in terms of introduction, study design, and presentation of results, but there are also some shortcomings.
1- The introduction provides the necessary background and some references, but it can be more in-depth and updated.
Answer: The introduction was briefly modified, as suggested.
2- The study design was reasonable and the methods were described in detail, however some details need to be improved. The results are clear and support the conclusions well, but the complexity of the results needs to be better explained in the discussion.
Answer: Adjustments were provided in all text, as pointed.
3- Overall, the articles have some academic value, but improvements in some aspects are needed to improve the quality and impact of the research. The current sample size on the findings should be discussed, such as whether the sample size may have prevented some effects from being detected or increased the risk of false negative results because the sample size was too small. Consider whether a sensitivity analysis of sample size is needed to assess the effect of variation in sample size on the stability of the study conclusions.
Answer: The statistical power of this study was limited (0.18) due to the restricted sample. This limitation arose from the inclusion of routine patients, who were selected based on their availability and specific clinical conditions. Consequently, we included animals of varying breeds, sizes and body conformation, introducing variability within each group. While this heterogeneity reflects the diversity typically encountered in clinical practice and enhances the external validity of the findings, it likely reduced the ability to detect statistically significant differences.
4- For some complex results, such as cases where the trend of some indicators at different time points is inconsistent or does not correspond to expectations, preliminary explanations or speculations should be provided in the results section. For example, when discussing the changes in vitamin C levels, combining the surgical stress response and oxidative stress mechanisms, explaining why the OSL-R group had higher vitamin C levels in the early postoperative period and the possible association of this change with other relevant indicators (such as oxidative stress indicators, inflammatory response indicators, etc.) helps readers better understand the significance of the results.
Answer: alteration of the data interpretation, considering the statistical power, was realized in the text, and the study limitation is explained in the text (Line 521-533).
Round 2
Reviewer 1 Report
Comments and Suggestions for Authors
The authors have incorporated the corrections that were commented on in the first revision round
Reviewer 2 Report
Comments and Suggestions for Authors
Having specified the absence of cortisol dosage but explaining the lack of cortisol-dependent effects and with the clarifying changes on the molecules that indicate metabolic stress, the paper is clearer and more correct